# Exploring ways to support patients with noncommunicable diseases: A pilot study in Nepal during the COVID-19 pandemic

**Hanako Iwashita**[1]☯*, **Rabina Shrestha**[2]☯, **Uday Narayan Yadav**[3,4], **Abha Shrestha**[5], **Deepa Makaju**[2], **Yuriko Harada**[1], **Gaku Masuda**[1], **Lal Rawal**[6,7,8], **Archana Shrestha**[9], **Biraj Karmacharya**[9], **Rajendra Koju**[10], **Haruka Sakamoto**[11], **Tomohiko Sugishita**[1,12]*

**1** Department of Hygiene and Public Health, Tokyo Women's Medical University, Tokyo, Japan, **2** Research and Development Division, Dhulikhel Hospital, Kathmandu University Hospital, Dhulikhel, Nepal, **3** National Centre for Epidemiology and Population Health, Research School of Population Health, The Australian National University, Canberra, ACT, Australia, **4** Centre for Primary Health Care and Equity, University of New South Wales, Sydney, NSW, Australia, **5** Department of Community Medicine, Dhulikhel Hospital, Kathmandu University School of Medical Sciences, Dhulikhel, Nepal, **6** School of Health, Medical and Applied Sciences, College of Science and Sustainability, Central Queensland University, Sydney Campus, Australia, **7** Appleton Institute, Physical Activity Research Group, Central Queensland University, Rockhampton, QLD, Australia, **8** Translational Health Research Institute, Western Sydney University, Sydney, NSW, Australia, **9** Department of Public Health and Community Programs, Dhulikhel Hospital, Kathmandu University School of Medical Sciences, Dhulikhel, Nepal, **10** Department of Internal Medicine, Dhulikhel Hospital, Kathmandu University School of Medical Sciences, Dhulikhel, Nepal, **11** Graduate School of Public Health, St Luke's International University, Tokyo, Japan, **12** Yakushima Onoaida Clinic, Kagoshima, Japan

☯ These authors contributed equally to this work.
* sugishita.tomohiko@twmu.ac.jp (TS); iwashita.hanako@twmu.ac.jp (HI)

**Data Availability Statement:** All relevant data are within the paper and its Supporting Information files.

## Abstract

Global healthcare systems have faced unprecedented strain due to the COVID-19 pandemic, with a profound impact on individuals with non-communicable diseases (NCDs), a scenario particularly pronounced in low-income countries like Nepal. This study aimed to understand the experiences of and challenges faced by patients with NCDs in Nepal during the pandemic, focusing on healthcare service availability and identifying factors affecting healthcare use, with the goal of being prepared for future emergencies. This study utilized a telephonic survey of 102 patients with NCDs and 10 qualitative interviews with healthcare providers in the Kavrepalanchok and Nuwakot districts of Nepal. We used mixed methods, with both qualitative and quantitative approaches. Specifically, multiple correspondence analysis, hierarchical cluster analysis, and classification tree analysis were used as exploratory methods. The study revealed that while 69.6% of the participants reported no difficulty in obtaining medication, other questions revealed that 58.8% experienced challenges in accessing routine medical care. Major barriers, such as fear of infection, unavailability of medicine in rural areas, and lack of transportation, were found through the qualitative interviews. Meanwhile, participants identified innovative strategies, such as telemedicine and community-based awareness programs, as potential facilitators for addressing barriers that arise during pandemic situations such as COVID-19. The COVID-19 pandemic exacerbated challenges in accessing healthcare services for patients with NCDs in Nepal. Our findings suggest the need to design and implement telemedicine services for patients with NCDs, as

**Funding:** This research was supported by Japan Agency for Medical Research and Development (AMED) Grant Number JP21jk0110020 (TS and HS) and HEIWA NAKAJIMA FOUNDATION (HI). The founders had no role in study design, data collection and analysis, decision to publish, or preparation of the manuscript. The authors received no specific funding for this work.

**Competing interests:** The authors have declared that no competing interests exist.

well as community-based programs that aim to improve health literacy, encourage healthy behavior, prevent development of NCDs, and ensure continuity of care during such crises, especially in countries with limited resources.

## Introduction

In early 2020, the COVID-19 pandemic was declared a global health emergency. The burden of COVID-19 was felt considerably even in countries with strong economic statuses and good healthcare systems [1]. Case fatalities due to the pandemic were especially high in those older than 65 years [2], and those with chronic noncommunicable diseases (NCDs), including diabetes and cardiovascular diseases [3]. People with NCDs were more vulnerable to becoming severely ill when they contracted COVID-19 [4]. In Italy, for example, one report stated that 96.2% of patients who had died in hospital had comorbidities, primarily NCDs such as hypertension and type 2 diabetes mellitus [5]. According to a cross-sectional study conducted by Amatya et al. in Nepal between 2020 and 2021, the most commonly present comorbidity was hypertension (43.7%), followed by diabetes (25.8%) [6].

Nepal, which has been undergoing a rapid demographic and epidemiological transition, is currently facing the double burden of combating NCDs and communicable diseases, in addition to COVID-19 [7]. In 2020, the first year of the pandemic, Nepal had the highest number of infections per million population in Southeast Asia, followed by the Maldives and India [8]. As of November 26, 2020, shortly after the survey used in this study was completed, 1,700,000 polymerase chain reaction (RT-PCR) COVID-19 tests had been conducted in seven provinces in Nepal, of which 227,640 were positive [9]. It was becoming clear that countries with increasing numbers of NCDs were highly vulnerable to COVID-19. The pooled prevalence estimates of hypertension increased from 26% to 32% between 2000–2005 and 2016–2020 [10], and the prevalence of type 2 diabetes mellitus increased from 8.4% to 8.5% between 2014 and 2020 [11]. Cases of coexistence of two and more NCDs were also reported [12].

Prevention and management services for patients with NCDs suffered severe disruptions since the start of the pandemic [13]. The World Health Organization (WHO) examined 155 countries and reported a serious impact on healthcare service delivery in high-, low-, and middle-income countries [14]. It stated that many people worldwide requiring treatment for NCDs on a regular basis had not been receiving healthcare services and essential medication owing to COVID-19 [15]. The same is true in Nepal, especially among older adults with pre-existing noncommunicable conditions [16]. The pandemic shifted the healthcare system's priorities to focus on COVID-19; consequently, health programs aimed at early detection of NCDs were scaled back, and potential patients with NCDs were overlooked [5]. The COVID-19 pandemic has affected many aspects of life for patients with NCDs, who have reported anxiety around managing their conditions as well as COVID-19 [13].

Although the WHO has already noted the importance of health systems in the fight against NCDs [15], more knowledge is needed to ensure that these systems are even stronger when emergencies such as the COVID-19 pandemic occur.

The COVID-19 pandemic began just as a diabetes prevention and control project was being launched in Nepal [17]. Even after the travel restrictions imposed during the COVID-19 pandemic were lifted, it was not possible to plan an actual community and health system visit immediately. However, there was an urgent need to investigate the challenges faced by people with NCDs, as they were more vulnerable to COVID-19 than the general population. As they

were constantly dealing with NCDs, it is possible that they were able to adopt behavioral changes that could help them deal with COVID-19. In emergency situations like the COVID-19 pandemic, where services cannot be provided by healthcare professionals, patients with NCDs and other patients unfortunately must manage their symptoms on their own. Moreover, evidence has also highlighted the synergistic or syndemic impact of COVID-19 on people with NCDs, which resulted from the interaction of socioecological and biological factors [18]. It is therefore important to know the extent to which health-related behaviors were or were not adopted during the COVID-19 pandemic, especially among patients with NCDs.

Health behavior is defined as the intentional activity of an individual for the purpose of managing an individual's health. [19]. The true effectiveness of health behavior is often unknown at first, but the patient may believe in its effectiveness, or the behavior change may have occurred unknowingly. First, regardless of its effectiveness, finding locally adaptable health behaviors in the field is a major contribution, and careful observational surveys would be useful in studying these behaviors. Therefore, this study aimed to understand the experiences of and challenges faced by patients with NCDs in Nepal during the pandemic, focusing on healthcare service availability and identifying factors affecting healthcare use.

## Materials and methods

### Ethics statement

Ethics approval was obtained from the Nepal Health Research Council Ethics Review Committee (Reg No. 944/2019) and the ethical committee at Tokyo Women's Medical University (No. 200801). The patients did not receive any reward for their participation. Verbal consent was obtained from each participant via telephone.

### Study setting

This study was conducted in Kavrepalanchok and Nuwakot, each with populations of 381,937 and 277,471 [20,21], respectively. Kavrepalanchok is located in the mid-hill region of Nepal, 30 km to the east of Kathmandu, the capital city. Nuwakot is situated in the hills. Both regions are largely populated by low-income subsistence farmers.

### Study design

To fulfil our aim, this cross-sectional pilot study was conducted through a mixed-methods approach following a convergent design [22]. It comprised two approaches: a quantitative approach, in which we collected data using questionnaires, and a qualitative approach, where data were collected through interviews with patients with NCDs and healthcare providers. Owing to travel restrictions, interviews with the participants were conducted via telephone.

Face-to-face surveys were not possible during the COVID-19 pandemic; this obliged researchers to implement remote surveys via telephone and/or other means. Even with limited survey methods, such as telephone surveys, we tried to understand as much as possible about the situation of people with NCDs in the area. We did not focus on implementing a flawless study design; rather, we aimed to explain carefully how we identified the conditions faced by people with NCDs during the COVID-19 pandemic. We hope this research will enable innovative approaches that will allow for urgent assessment of a patient's situation in an emergency and immediate assistance even in remote locations.

## Study population

The study included patients who were older than 18 years of age and who had been diagnosed as having chronic NCDs (including diabetes, hypertension, cardiovascular disease, chronic obstructive pulmonary disease [COPD], orthopedic diseases, hypothyroidism, and kidney disease). Patients were mainly those using the Dhulikhel Hospital in Kavrepalanchok and its outreach centers and governmental healthcare facilities in Kavrepalanchok and Nuwakot. Healthcare providers and policymakers comprised those affiliated with facilities in the same districts.

## Sampling technique

Using a convenience sampling strategy, quantitative data were collected from 102 patients with NCDs by obtaining a list of patients and their contact numbers from the government health centers and some outreach centers of Dhulikhel Hospital. Qualitative data were collected from 18 of the 102 patients—who were also included in the quantitative study—and 10 healthcare providers (health post in-charge, health coordinator) who were selected from the Kavre and Nuwakot districts. One central-level policymaker from the NCD section at the Department of Health Division, Nepal, was also interviewed using a qualitative approach. Regarding patients, there was overlap in the interviewees for the qualitative and quantitative approaches, while regarding non-patients, the interviewees were addressed using the qualitative approach only. The recruitment period for this study was September 1 to October 15, 2020, during the COVID-19 pandemic, and data collection was conducted during the same period.

## Data collection

Quantitative data were collected using questionnaires covering comprehensive topics, including baseline information, self-perceived difficulty, precautionary behavior toward COVID-19, psychological impacts caused by the COVID-19 pandemic, and lifestyle behavior changes.

Qualitative data were collected using an interview guide (S1 Text). The members of the research team conducted the interviews in Nepali until the point of data saturation. All interviews were audio recorded and transcribed in Nepali. They were then translated into English by a research team member.

## Data analysis

In the quantitative data analysis as exploratory method, we performed a multiple correspondence analysis (MCA) [23,24], a hierarchical cluster analysis (HCA) [23] and a classification tree analysis (CTA) [25]. Statistical software R v.4.1.1 was used to analyze the data (the details are in S2 Text). In the qualitative data analysis, the first and second authors developed the first set of codes with the iteration of the code book. These codes were revised by the other authors and checked for inter-coder reliability. Themes were developed from the codes generated through the qualitative study findings [26]. Finally, triangulation of quantitative and qualitative survey results was performed.

# Results

## Baseline information

To understand the situation of patients with NCDs and assess the accessibility of healthcare services during the COVID-19 pandemic, we conducted a comprehensive survey involving 102 participants. The mean age (±standard deviation [SD]) was 56.1(±12.7) years. Men

constituted 61.8% (63/102) of all the participants. Table 1 presents the details of the questionnaire. Most patients with NCDs had diabetes alone (41.2%, 42/102), diabetes and hypertension (25.5%, 26/102), hypertension alone (12.7%, 13/102), or other patterns (such as Thyroid disorder and Hydronephrosis, etc.; 20.6%, 21/102). The median period since diabetes and hypertension were diagnosed were 6.2(±5.8) and 7.4(±7.7) years, respectively. The variant depended on the age ranges. Although it was expected that the duration of the disease would be higher in people older than 60 years, there was no particular trend by age. At the time of the survey, two patients (2/102) were diagnosed with COVID-19. Of them, one patient had hypertension only and the other had diabetes and hypertension.

### Situation of patients with NCDs during the COVID-19 pandemic

Examining the outcomes of our questionnaire survey, we observed that a majority of the participants (69.6%, 71 of 102) reported no difficulty in obtaining their medication during the pandemic period (refer to B-2 in Table 1 for more details). We asked about the status of service delivery owing to the COVID-19 pandemic. The quotes below are from the qualitative results, and the symbols used for anonymity indicate whether they were patients (e.g., D002, Th001, CVD002, D001, D006, Neph001, R003, D003, D004) or healthcare providers (e.g., RMC1, RMC2, UMC2, N1, N2). Moreover, for patients, the NCD(s) they had been diagnosed with are also indicated below.

The responses of health coordinators from different municipalities varied. Specifically, some stated that they stopped the service while others stated that they had not stopped the service and their health facilities were open.

The following responses were received from healthcare providers:

*"We have not stopped any services."* (RMC2, Health coordinator)

*"Health facilities are open but the services are totally stopped now."* (RMC1, Health coordinator)

A patient said that they had no trouble procuring medicines during this period:

*"Medicine is available in clinic."* (D002, Patient, Diabetes and Hypertension)

*"I haven't felt any differences. When I went to Kharanitar* (Place name) *the doctor was available at that hospital; so, it was easy. I am receiving services."* (D002, Patient, Diabetes and Hypertension)

Another patient said:

*"I haven't faced any difficulties till now. Those who have to go out couldn't go. People like us stay at home. We haven't been out."* (CVD002, Patient, Hypertension and COPD)

Approximately half of the patients (58.8%: 60/102) reported having some difficulty obtaining routine medical care (B-3 in Table 1). The questionnaire results showed that 20.6% (21/102) of the patients cancelled their doctor's appointment (C-5 in Table 1), but only one relied on alternative routes like telehealth/telemedicine (C-5 and C-12 in Table 1). In this case, telehealth/telemedicine refers to consultations with doctors or nurses via the internet or telephone. Even though medication was available, there were cases in which patients encountered difficulty adjusting their dosage without consulting a doctor. A patient said:

**Table 1. Questionnaire for all patients.**

| Question | Answer | No. of patients | Category names for MCA | No. of patients | Abbreviation (for MCA) |
|---|---|---|---|---|---|
| A. Baseline information | | | | | |
| A-1. Age | Above 60 years | 32 | Above 60 years | 32 | ag_a |
| | 60 years or less | 70 | 60 years or less | 70 | ag_l |
| A-2. Gender | Female | 39 | Female | 39 | f |
| | Male | 63 | Male | 63 | m |
| A-3. Ethnicity [1] | Dalit | 1 | Others | 36 | et_n |
| | Disadvantaged Janajati | 6 | | | |
| | Disadvantaged non-Dalit Terai caste groups | 1 | | | |
| | Religious minorities | 1 | | | |
| | Relatively advantaged Janajati | 27 | | | |
| | Upper caste groups | 66 | Upper | 66 | et_y |
| A-4. Religion | Hindu | 95 | Hindu | 95 | re_h |
| | Buddhist | 5 | Others | 7 | re_o |
| | Muslim | 0 | | | |
| | Christian | 1 | | | |
| | Others | 1 | | | |
| A-5. Education qualification | Higher education | 7 | Higher and Secondary | 21 | ed_h |
| | Secondary | 14 | | | |
| | Primary | 35 | Primary, Able, and Illiterate | 81 | ed_p |
| | Able | 23 | | | |
| | Illiterate | 23 | | | |
| A-6. Occupation | Professional | 8 | Others | 73 | oc_o |
| | Technical | 6 | | | |
| | Managerial Clerical Sales | 3 | | | |
| | Skilled | 9 | | | |
| | Unskilled | 47 | | | |
| | Agriculture | 29 | Agriculture | 29 | oc_a |
| A-7. Diabetes | Diabetes_Yes | 81 | Diabetes_Yes | 81 | Di_d |
| | Diabetes_No | 21 | Diabetes_No | 21 | Di_n |
| A-8. Hypertension | Hypertension_Yes | 51 | Hypertension_Yes | 51 | Hy_h |
| | Hypertension_No | 51 | Hypertension_No | 51 | Hy_n |
| B. Self-perceived difficulty | | | | | |
| B-1. Are you currently taking medicines for chronic conditions? | Yes | 94 | Not used in MCA | | |
| | No | 8 | Not used in MCA | | |
| B-2. How difficult was it for you to obtain your medication owing to the COVID-19 pandemic or social distancing rules? | Unable or very difficult | 3 | Yes | 31 | dm_y |
| | Very | 14 | | | |
| | Slightly | 14 | | | |
| | No | 63 | No | 71 | dm_n |
| | Answered "No" to question B-1 | 8 | | | |
| B-3. How difficult was it for you to access routine medical care that you need owing to the COVID-19 pandemic or social distancing rules? | Difficulty | 17 | Yes | 60 | dc_y |
| | Very | 15 | | | |
| | Slightly | 28 | | | |
| | None | 42 | No | 42 | dc_n |

(*Continued*)

**Table 1.** (Continued)

| Question | Answer | No. of patients | Category names for MCA | No. of patients | Abbreviation (for MCA) |
|---|---|---|---|---|---|
| C. Precautionary behavior regarding COVID-19 Which of the following actions have you taken in the last several days to keep yourself safe from COVID-19 (in addition to those you normally do)? (Select all that apply) | | | | | |
| C-1. Worn a face mask | Yes | 92 | Yes | 92 | Fa_y |
| | No | 10 | No | 10 | Fa_n |
| C-2. Washed/sanitized hands | Yes | 92 | Yes | 92 | Sa_y |
| | No | 10 | No | 10 | Sa_n |
| C-3. Avoided public places/ crowds | Yes | 88 | Yes | 88 | Ac_y |
| | No | 14 | No | 14 | Ac_n |
| C-4. Avoided in-person contact with high-risk people | Yes | 28 | Yes | 28 | Ap_y |
| | No | 74 | No | 74 | Ap_n |
| C-5. Canceled a doctor's appointment | Yes | 21 | Yes | 21 | Cd_y |
| | No | 81 | No | 81 | Cd_n |
| C-6. Avoided in-person contact with friends or family | Yes | 19 | Yes | 19 | Af_y |
| | No | 83 | No | 83 | Af_n |
| C-7. Canceled/postponed travel | Yes | 13 | Yes | 13 | CT_y |
| | No | 89 | No | 89 | CT_n |
| C-8. Stockpiled food/water | Yes | 12 | Yes | 12 | St_y |
| | No | 90 | No | 90 | St_n |
| C-9. Prayed | Yes | 10 | Yes | 10 | Pr_y |
| | No | 92 | No | 92 | Pr_n |
| C-10. Visited a doctor in person | Yes | 9 | Yes | 9 | Vi_y |
| | No | 93 | No | 93 | Vi_n |
| C-11. Isolated myself from others who live with me | Yes | 8 | Yes | 8 | Is_y |
| | No | 94 | No | 94 | Is_n |
| C-12. Had a "telehealth visit" with a doctor or other healthcare provider | Yes | 8 | Yes | 8 | Te_y |
| | No | 94 | No | 94 | Te_n |
| C-13. Worked or studied at home | Yes | 5 | Yes | 5 | Ho_y |
| | No | 97 | No | 97 | Ho_n |
| C-14. Canceled/postponed work or school activities | Yes | 4 | Yes | 4 | Ca_y |
| | No | 98 | No | 98 | Ca_n |
| D. Psychological impact of the COVID-19 pandemic | | | | | |
| D-1. How concerned are you about the COVID-19 pandemic? | Very concerned | 25 | Yes | 60 | co_y |
| | Somewhat | 35 | | | |
| | Not at all | 42 | No | 42 | co_nt |
| D-2. While watching the news and listening to stories on COVID-19 via social media, I become nervous or anxious. | Strongly agree | 14 | Agree | 38 | ne_a |
| | Agree | 24 | | | |
| | Neither agree nor disagree | 29 | Not agree (Included Neither agree nor disagree) | 64 | ne_n |
| | Disagree | 31 | | | |
| | Strongly disagree | 4 | | | |
| E. Lifestyle behavior changes | | | | | |
| E-1. Compared with the months before the outbreak began, how frequently did your communication with close friends and family change? | I communicate with them less often than before. | 64 | Less | 64 | fr_l |
| | I communicate with them about the same as before. | 30 | Not less | 38 | fr_n |
| | I communicate with them more often than before. | 8 | | | |

(*Continued*)

**Table 1.** (Continued)

| Question | Answer | No. of patients | Category names for MCA | No. of patients | Abbreviation (for MCA) |
|---|---|---|---|---|---|
| E-2. How often has your sleep been interrupted or disturbed because of your concerns about the outbreak of COVID-19? | Not at all | 86 | No | 86 | sl_n |
| | Somewhat | 11 | Yes | 16 | sl_y |
| | A lot | 5 | | | |
| E-3. Do you smoke? | Yes | 13 | Yes | 13 | sm_y |
| | No | 89 | No | 89 | sm_n |
| E-4. Do you drink alcohol? | Yes | 10 | Yes | 10 | dr_y |
| | No | 92 | No | 92 | dr_n |
| E-5. Before this outbreak, were you doing physical (light or rigorous) activity? | Yes | 57 | Yes | 57 | ph_y |
| | No | 45 | No | 45 | ph_n |

[1] The items are similar to those in the STEPS survey.

https://www.researchgate.net/publication/262830102_Non_Communicable_Diseases_Risk_Factors_STEPS_Survey_Nepal_2013.

*"We don't know if the dose of medicine should be increased. We have been taking medicine regularly."* (Th001, Patient, Diabetes and Thyroid disorder)

We used MCA and HCA to understand better the circumstances of patients with NCDs (Fig 1). Through HCA, we identified three distinct groups or clusters of patients, each characterized by unique attributes. The first group, Cluster 1, primarily comprised patients who encountered minimal challenges in managing their NCDs. These patients exhibited lesser psychological impact due to the pandemic (refer to S1 Fig, S1 Table for additional insights). Cluster 2 is located at the lower left of Fig 1. Most patients (25/39, 90%) categorized as Cluster_2 encountered some difficulty in managing their NCDs, which made them a psychologically susceptible group. However, their responses regarding "Precautionary behaviors" were not particularly distinctive, with most patients showing the same tendency to answer similarly as the population in this study. Cluster 3 is located at the right side of Fig 1, comprising only seven patients, and is characterized by a tendency to answer "Yes" to the question regarding "isolation" (Is_y), which highly contributes to Dimension 1. In Cluster 3, not only "isolation" (Is_y), but also precautionary behaviors, such as "cancelling/postponing work or school activities" (Ca_y) and "visiting a doctor in person" (Vi_y), were excellent. Although they isolated themselves properly, their frequency of communication remained unchanged. Notably, this is an ideal behavior observed by chance, even during a pandemic. While the trends regarding "Precautionary behavior toward COVID-19" and "Lifestyle behavior changes" among these clusters differed, only Cluster 3 was characterized by a strong positive aspect and seemed to have advantages such as innovative approaches on how to make continuous service provision possible.

## Potential barriers for service utilization

1) People's fear of getting infected by health workers

One barrier was the fear of contracting COVID-19 from infected health workers; patients considered health centers as the biggest source of infection.

*"Now, everyone is scared to go to the hospital. The more the health workers get infected, the more difficult it is for others. That is why I am not going anywhere."* (D001, Patient, Diabetes)

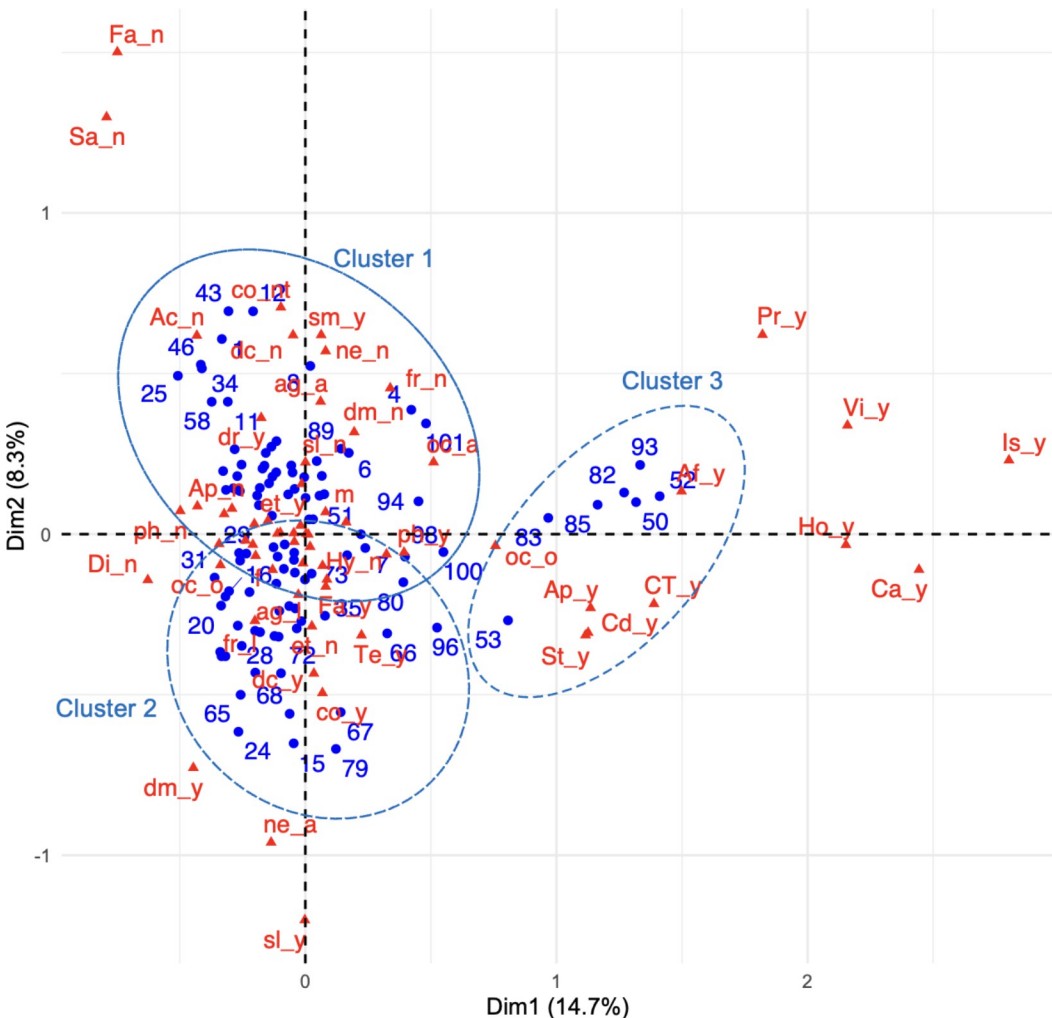

**Fig 1. MCA biplot positioning of patients and variables, vis-à-vis both dimensions.** Small blue circles represent the patients. Red triangles represent the variables. The variable names are as shown on the right side of Table 1. Large dotted circles indicate clusters.

2) Unavailability of medicine in rural areas: One patient addressed the issue of differences in the availability of medication between rural and urban areas, and focused on awareness activities in rural areas:

*"Because of the lockdown and having no access to transportation, I could not do follow-ups properly. At the moment, with the help of an ambulance driver, I have been taking the medicine. Now, for the dose adjustment, I need to visit a hospital and consult with a doctor for the proper dose of my medicine, if any change is needed."* (R002, patient of respiratory problem)

3) Lack of availability of transportation: Regardless of the availability of medical services, the lack of accessible transportation to the health center presented a barrier.

*"Yeah, yeah, transportation is a problem mainly for people living in villages, especially during this COVID crisis. Those who live in urban areas don't have problems. We haven't had any difficulties until now. Others might be facing difficulties."* (D001, Patient, Diabetes)

*"Yeah, there is a problem with transportation. I have my own bike at home, so my son takes me for checkup. Previously, I used to go in an ambulance. We have ambulance in our committee."* (D001, Patient, Diabetes)

*"There is difficulty. I used to go for my checkup at Dhulikhel. During this time, it is difficult to go to the hospital."* (Th001, Patient, Diabetes and Thyroid disorder)

Another patient expressed a similar sentiment,

*"As I said before, it is very difficult to perform a polysomnography test. If the COVID-19 crisis hadn't occurred, I would have been able to do the test within the past 2–3 months. I would have been healthier. I would have been aware of my situation early; I wasn't able to know about my condition in time."* (R002, COPD patient)

4) Less follow up than usual: For patients with diseases like hypertension and diabetes, and even other chronic diseases, dose adjustments are required, followed by regular follow-ups, but the situation led to problems in following-up at the hospital for investigations and dose adjustments.

This sentiment was verbalized by participants as follows:

*"I thought of going for checkup but because of this lockdown, it is difficult to go,"* and

*"It's been 4/5 months that I haven't tested."* (D003, patients with diabetes)

5) Lack of opportunity to take the polysomnography test: Some of the patients with respiratory disorder were most at risk of misdiagnosis because of the similarities in symptoms with COVID-19. For example, it was not possible to take a polysomnography test at the health center because of the risks of contracting the virus from the staff. This was expressed by one of the COPD patients as follows:

*"I have learned two things from this COVID-19 situation: the first is we cannot do anything by staying at home, and the second is no hospital would admit you after you are infected with the virus. Every hospital gives you reasons like the staffs are sick, and the machine is not working. I informed them of my [financial] status, but even having money is useless. That day, I wasn't able to take the polysomnography test. I was affected by the COVID-19 situation and not able to take the test. In case I am not able to take the test in time and an emergency occurs while I stay in the village, I fear that I will die. That affected me deeply."* (R003, COPD patient)

## Facilitators of the use of healthcare

1) The use of telemedicine has noticeably increased after the COVID-19 pandemic: In a crisis situation, it seemed that people preferred to use telemedicine rather than visit a health center, as expressed by a patient regarding the possibility of providing counseling via telephone or social media, which was equally supported by the consulting doctor.

Comments regarding "telemedicine" were as follows:

*"Telemedicine would be good. We couldn't go [to the hospital], but if we could get advice, that would be very good. Doctors providing advice and patients receiving that advice is a very good thing."* (Neph001, Patient, Diabetes and Hydronephrosis)

"*That is good. Information must be provided about that kind of service. I don't know about that.*"

"*That is a good thing. People must be informed about this service. Most of the people don't know about it.*" (D001, Patient, Diabetes)

One patient who conditionally supported telemedicine commented that:

"*Normally, telemedicine is also a good concept, but for those who go to hospital on a regular basis, it may not be effective. It may be supportive only for urgent cases and beneficial for those who cannot have regular follow-ups because they live far from a hospital and don't have a health facility nearby. If we share our problem, doctors might offer some advice. Their small advice might be very helpful for us; so, I support this type of program.*" (R003, Patient, COPD)

"*Before COVID-19, I used to go to a medical [facility] or hospital immediately after I faced any health problem, but now, I generally consult with doctors by phone; most of my friends are doctors*" (CVD001, Patient, Diabetes, Hypertension, Cardiovascular Stroke).

"*Yeah, I have been consulting and I did ECG (electrocardiogram) and ECHO (Echocardiogram) as per consultation. I showed my report on Viber (telephone) and he told me that everything is okay and to continue taking my regular medicine. I have been doing so. He has been urging me to reduce my weight.*" (CVD001, Patient, Diabetes, Hypertension, Cardiovascular Stroke)

One of the health posts in charge also mentioned,

"*We have provided [our] office number and they will ask me.*" (K1, Health post in charge)

Another patient thought that telemedicine, including the prescription of medicines, needed to be developed, and noted that the procurement of medicines was an issue.

"*Even though we have done phone consultations; we need to go [to the hospital somehow] to take medicine. Because of that, people are terrified.*" (D004, Patient, Diabetes)

2) Awareness programs through mothers' groups, knowledgeable intellectuals, shopkeepers, dhami-jhakri, and lamas (traditional healers): During that time of crisis, health workers were quite busy with their works, which prompted the health authorities of municipalities to engage people from the community, like mothers' groups, intellectuals who are knowledgeable, shopkeepers, dhami-jhakri, and lamas, to help promote awareness, which was a good step toward a faster delivery of messages. Facilitating factors included the dissemination of knowledge and the opportunity to share information. One health coordinator from an urban municipality stated,

"*We have also organized awareness programs about the symptoms and preventive measures against COVID-19 in villages and in mothers' groups.*" (UMC2, Health coordinator)

A similar response was given by the health post in charge,

"*We have a team comprising ward members, women health volunteers, knowledgeable intellectuals, shopkeepers, dhami-jhakri, and lamas. We have discussed COVID-19 with them,*

*including its symptoms and preventive measures. They go around to conduct awareness [programs] while practicing social distancing and wearing masks. They also talk about hygiene.*"
(N1, Health post in-charge)

"*Awareness is necessary. Most of the people may not know; some may be suffering from diseases. So, first of all, spreading awareness is necessary by going to every ward.*" (D001, Patient, Diabetes)

## Useful field information obtained through exploratory methods

In this study, exploratory methods were used to obtain information about the situation in the field during the COVID-19 pandemic. A total of 32 variables from the questionnaire options, namely, 8 basic (age, gender, ethnicity, religion, education, occupation, history of diabetes and hypertension) (A in Table 1) and 24 explanatory variables; 3 for "Self-perceived difficulty" (B in Table 1), 14 for "Precautionary behavior toward COVID-19") (C in Table 1), 2 for "Psychological impact of the COVID-19 pandemic" (D in Table 1), and 5 for "Lifestyle behavior changes" (E in Table 1), were drawn. Using CTA, a combined analysis was conducted with one of the 32 variables as the response variable and the remaining variables as explanatory variables. We found notable results regarding nervousness and anxiety about the pandemic (Fig 2). Of the included participants, 37.3% (38/102) were nervous or anxious because they

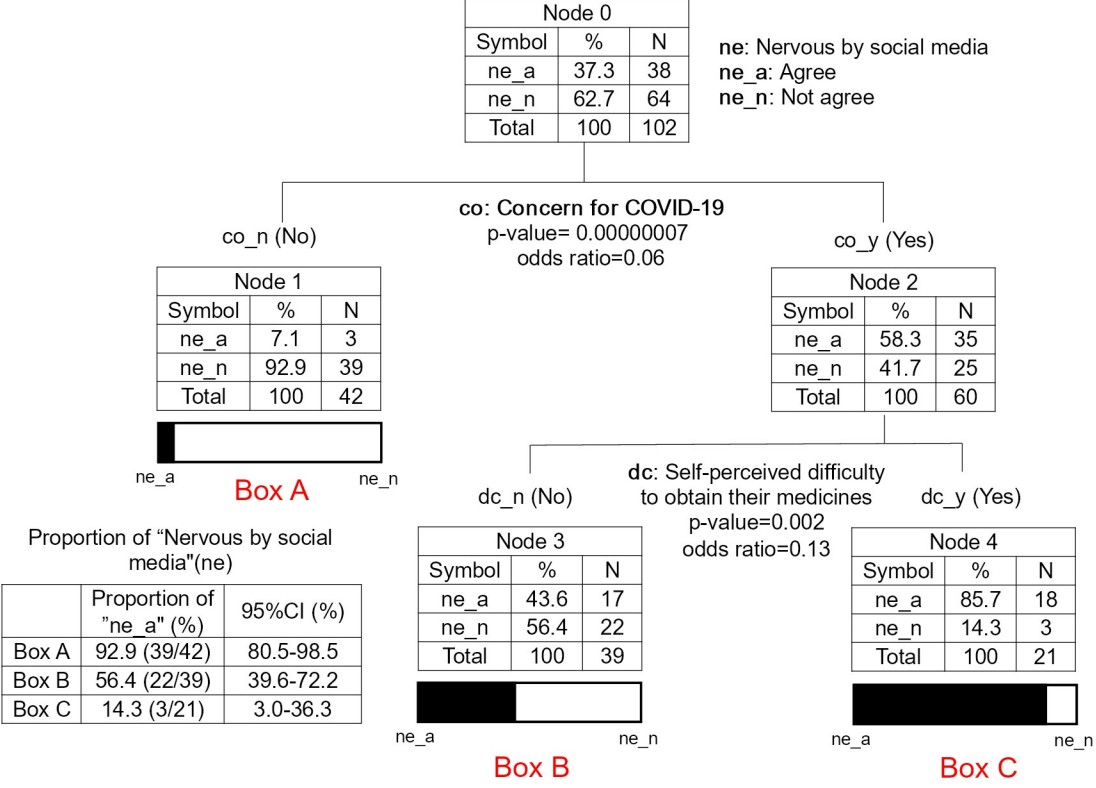

The 95% confidence interval (CI) for the proportion of objective variables in each subgroup was calculated using the Clopper–Pearson method.

**Fig 2. CTA shows sub-typing of patients with different feelings, namely nervousness and anxiety owing to social media.** Note: This tree includes only the variables that contribute significantly to sub-typing patients in terms of the number of patients who felt anxious because of the news on COVID-19 on social media.

saw news related to COVID-19 on social media (D-2 in Table 1, Fig 2). First, the CTA explained the trend: for the 92.9% (95% CI: 80.5, 98.5%) of patients who were not concerned about the COVID-19 pandemic, the news on social media regarding COVID-19 did not make them nervous or anxious (Fig 2, Box A). Second, among patients who were concerned, the proportion of those who felt nervous or anxious because of the news on COVID-19 on social media varied according to whether they encountered difficulties in obtaining medication (Fig 2, Boxes B and C). They were more likely to feel nervous or anxious because of such news if they felt they had no access to medication. Hence, efforts should be made to ensure that social media facilitate access to medicine for patients with NCDs who encounter difficulty in obtaining their medicine; for example, by including information on drug procurement in a convenient manner.

## Discussion

In emergency situations, such as the COVID-19 pandemic, it is often difficult to conduct surveys. Encountering this situation required us to adopt an imperfect survey approach to avoid face-to-face surveys. During the study period, the status of healthcare service delivery changed daily and varied from facility to facility. However, despite the fact that this was a cross-sectional survey with a limited tool, that is, the telephone, we were able to gather the opinions of patients with NCDs who were willing to communicate with us.

To understand the situation of patients with NCDs and the availability of healthcare services, various combinations of questions and interviews were carefully explored through descriptive analysis. Furthermore, MCA and HCA methodology allowed us to identify an ideal patient population (Cluster 3) in which proper countermeasures appeared to have been undertaken (S1 Table). We found that MCA and HCA were useful for quickly finding relationships and structures with limited data from telephone interviews in situations where face-to-face contact is avoided. A more in-depth, qualitative investigation of this Cluster 3 would have uncovered suggestive behaviors, however, this was unfortunately not possible because direct interviews and observations were avoided. Although the qualitative approach indicated that healthcare and drug provision services were barely normal during the study period, we found that each patient had their own individual problems. For example, even if medicine could be procured, if no doctor was available, the appropriate prescription for the medicine could not be determined. More in-depth interviews with Cluster 3 patients may have provided some insight into how to respond to each patient's specific problems in an emergency situation.

Qualitative research identified the barriers and facilitators for the use of NCD services. In any case, we received a wide range of opinions that were indistinguishable as to the time of their formulation, that is, before or during the COVID-19 experience in Nepal. In fact, some of the issues discussed in this study may have merely exposed problems that had existed in Nepal long before COVID-19 spread. However, the COVID-19 pandemic made the situation more challenging in Nepal, where NCDs are difficult to manage successfully, and people showed varied forms of anxiety. It is possible that anxiety about not being able to receive medical care from a doctor or access medicines, including transportation issues, could be addressed through daily preparation, by gaining knowledge on NCDs and the relationships among professionals and patients. It is also important to be able to respond remotely, for example, through telemedicine, when immediate help is needed. In this study, only one of the patients who canceled their visit with their doctor had access to telemedicine; hopefully, in the future, all patients will have this option.

During our investigation, we observed that a significant proportion of patients with NCDs encountered obstacles in obtaining regular medical treatment throughout the COVID-19

pandemic. Additionally, these experiences had substantial psychological impacts on these individuals. Despite these challenges, our study highlights potential mitigating strategies, including the implementation of telemedicine and community-driven awareness campaigns. Telemedicine, defined as the provision of healthcare services remotely through the utilization of digital platforms and technology by healthcare professionals [24], manifested as a promising solution during the COVID-19 pandemic. This approach, combined with community-based initiatives, may represent a transformative means to alleviate the difficulties faced by patients with NCDs in these unprecedented times. Indeed, some patients with NCDs feared hospital transmission of COVID-19; hence, self-management of NCDs remained inadequate. Williamson et al. [25] indicated that people with diabetes were up to three times more likely to have severe symptoms or die from COVID-19. The situation is likely worse for those with uncontrolled diabetes [27]. Patients with hypertension were 3.5 times more likely to die of COVID-19 [28]. Therefore, to reduce the burdens of high risk of COVID-19 for patients with NCDs, well-established strategies to improve treatment are crucial for them to manage their condition. Telemedicine and home delivery of medication can be encouraged by doctors and pharmacists, if a patient finds it difficult to procure their medication or regular prescriptions [29]. This approach may be useful especially for patients who are afraid of contracting COVID-19 through hospital visits. Patients were exposed to this risk and encountered various challenges in obtaining medication and healthcare. For example, even when medicine was available during an emergency situation, it was difficult to ascertain if the correct dosage was being taken, as patients often did not have a prescription from a doctor.

Globally, when considering both low- and high-income countries, the COVID-19 pandemic caused a general decline in health service use and exposed a variety of barriers to service use, even for routine treatment of NCD [30]. Therefore, strategies for the prevention of NCDs should be routinely implemented in daily life [31]. A NCD monitoring system can help people with NCDs manage their condition themselves [32]. This study suggests that the implementation of awareness programs run by culturally aligned local groups is effective in actually enhancing crisis response and management. These groups could help to effectively convey various awareness-related messages on behalf of health workers who are busy during emergencies. Regardless of the topic of the programs, NCDs or otherwise, the key is to conduct such activities during normal times; therefore, when there is an emergency situation that requires avoiding face-to-face contact, telemedicine and other services can be used to facilitate appropriate connections with professionals.

Social media encompass tools and platforms for social interaction such as digital and web-based and mobile technologies [33]. In situations such as the COVID-19 pandemic, social media has great potential [34]. However, social media should be used in such a way that patients with NCDs do not feel uncomfortable. According to the questionnaire survey results, approximately one-third of the respondents felt nervous or anxious upon hearing news on COVID-19 on social media. This study showed that negative feelings evoked by social media were common in patients who were concerned about COVID-19 and encountered difficulty in obtaining their medicine (Fig 2, "Nervous by social media" response is more common in Box C than in Box B). People on social media should avoid triggering such negative feelings in patients and instead focus on communicating facts, such as including information on the procurement of medication.

The use of social media and alternative communication channels has facilitated global information dissemination, even reaching low-income countries. However, this accessibility has concurrently created the potential for misunderstanding [35]. In this context, it is paramount to meticulously validate the accuracy and reliability of all information received. In most emergency situations, multiple sources of information could be simultaneously exploited, and

recipients may get confused or be mistaken about the accuracy of a given piece of information [36]. In some cases, information may be extremely limited [37]. Providing unreliable or imprecise information can cause more harm such as overwhelming fear and information overload than good [38]. This is dangerous because incorrect information, that may distress patients, spreads faster than COVID-19 [39]. Once feelings of nervousness or anxiety are triggered in patients with NCDs, they may not be able to distinguish between imprecise and precise information [40]. To avoid this situation, it is necessary to verify whether the given information is from a reliable source [41].

## Implications

As one innovative approach derived through our aim, we introduce a modified "positive deviance approach (PDA)" that could be incorporated into the methodology of this study. This approach involves identifying contextual health behaviors in Nepal, which may be disguised as "deviant behavior," and disseminating them to the local community. It could be integrated into health promotion materials to influence collaborative changes with community groups [42]. Based on PDA, a "positive deviance group (PDG)," which comprises "positive deviants (PDs)" could be formed. These PDs could identify behaviors that enable PDGs or individuals to overcome a problem, even without special resources. Although PDA normally requires a deep connection with a given community, and begins with in-depth observation activities, we found it difficult to conduct face-to-face surveys during the COVID-19 pandemic; therefore, we recommend a new method that could be done remotely. We used mixed methods with qualitative and quantitative approaches, as well as exploratory methods such as MCA and HCA. An important concept of this approach is to find wisdom-based behaviors that lead to solutions for people with similar socio-demographic characteristics in a community. In fact, such behaviors are often overlooked when viewed through the lens of traditional norms, although the community might have already succeeded in discovering them. In this sense, the input of outsiders might be useful in finding the ideal PDG. In this study, the use of MCA and HCA allowed us to identify patients who were classified in Cluster 3; our encounter with Cluster 3 was reminiscent of the PDA.

## Limitations

While our study provides important insights, it is important to acknowledge certain limitations. First, as convenience sampling from only two districts was used, there may be concerns regarding the generalizability of the findings. In addition, differences between the two regions (Kavrepalanchok and Nuwakot) were not taken into account in the analysis. This approach is susceptible to sampling bias, as it relies on the availability of participants and may not accurately represent the broader population of patients with NCDs. Second, the reliance on telephone-based surveys does not reflect the views of those who did not have enough time to talk on the telephone, even though there were no telephone charges for the patients. Such methods run the risk of self-selection bias, which may render the results less interpretable than those obtained from surveys based on random sampling [43]. Third, we targeted participants from a list of patients with NCDs from specific hospitals or healthcare centers only in Kavrepalanchok and Nuwakot. The results may be limited in generalizability to other areas in Nepal. Fourth, we could not control for social-desirability bias, which may have led the participants to provide answers that would make them look good to the researcher [44]. Holbrook et al. indicated that social desirability bias might occur more frequently in telephone interviews compared to face-to-face interviews [45]. On the other hand, because it was a telephone survey, participants were able to answer more honestly, in some cases [46], especially in emergencies, such as the

COVID-19 pandemic. In face-to-face interviews, eye contact may be intimidating for some participants [47]. Finally, owing to the cross-sectional study design, the temporality between variables was unclear.

## Conclusions

Our findings from the study provide a valuable insight into the challenges and experiences of patients with NCDs during the COVID-19 pandemic. In addressing our aim, we found that nationwide lockdowns, social distancing, and travel restrictions should be adopted. Under such circumstances, it is important for all patients, including those with NCDs, to know how to respond to and manage crises on their own from the early stages.

To enhance the crisis response and management, the implementation of awareness programs by culturally aligned local groups can be impactful. These groups could effectively convey various awareness-related messages on behalf of health workers who are busy with their work during the emergency. Whatever the topic is, NCDs or not, the key is to conduct such activities during normal times, and finally, when there is a need to avoid face-to-face contact in a true emergency situation, telemedicine and other services could be used to facilitate appropriate connections with professionals.

While the approach outlined above is inherently promising, there is a possibility of missing pertinent information. Hence, our data mining approach will likely uncover distinctive and ideal groups that could offer further insights. To connect to useful measures in the larger community, it would be worthwhile to take this survey method one step further and use various approaches, including qualitative and quantitative ones.

## Supporting information

**S1 Text. Interview guide.**
(DOCX)

**S2 Text. Data analysis.**
(DOCX)

**S1 Fig. Factor map of the hierarchical cluster analysis.**
(TIFF)

**S1 Table. Differences in responses in each cluster.**
(DOCX)

## Acknowledgments

We are deeply grateful to the patients and health providers in Kavrepalanchok and Nuwakot who participated in this study. Our heartfelt appreciation goes to Dr. Rajeev Shrestha, Dr. Osamu Kiritani, Mr. Daisuke Kosugi, Ms. Yuki Yoshino, Ms. Airi Furuya and Ms. Hanae Hata for providing administrative support.

## Author Contributions

**Conceptualization:** Hanako Iwashita, Rabina Shrestha, Lal Rawal, Tomohiko Sugishita.

**Data curation:** Hanako Iwashita, Rabina Shrestha, Uday Narayan Yadav, Abha Shrestha, Deepa Makaju, Yuriko Harada, Gaku Masuda, Lal Rawal, Archana Shrestha, Biraj Karmacharya, Rajendra Koju, Haruka Sakamoto, Tomohiko Sugishita.

**Formal analysis:** Hanako Iwashita, Rabina Shrestha, Uday Narayan Yadav, Gaku Masuda.

**Funding acquisition:** Haruka Sakamoto, Tomohiko Sugishita.

**Methodology:** Hanako Iwashita, Rabina Shrestha, Uday Narayan Yadav, Abha Shrestha, Deepa Makaju, Yuriko Harada, Gaku Masuda.

**Project administration:** Hanako Iwashita, Rabina Shrestha, Abha Shrestha, Deepa Makaju, Yuriko Harada, Lal Rawal, Archana Shrestha, Biraj Karmacharya, Rajendra Koju, Haruka Sakamoto, Tomohiko Sugishita.

**Supervision:** Uday Narayan Yadav, Rajendra Koju, Haruka Sakamoto, Tomohiko Sugishita.

**Writing – original draft:** Hanako Iwashita.

**Writing – review & editing:** Rabina Shrestha, Uday Narayan Yadav, Abha Shrestha, Deepa Makaju, Yuriko Harada, Gaku Masuda, Lal Rawal, Archana Shrestha, Biraj Karmacharya, Rajendra Koju, Haruka Sakamoto, Tomohiko Sugishita.

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
