## [Decision Letter · Decision Letter 0]

15 Feb 2024

PGPH-D-23-02128

Exploring ways to support patients with noncommunicable diseases: a pilot study in Nepal during the COVID-19 pandemic

Dear Dr. Iwashita,

Thank you for submitting your manuscript to PLOS Global Public Health. After careful consideration, we feel that it has merit but does not fully meet PLOS Global Public Health’s publication criteria as it currently stands. Therefore, we invite you to submit a revised version of the manuscript that addresses the points raised during the review process.

We look forward to receiving your revised manuscript.

Kind regards,

Sok King Ong

Academic Editor

Journal Requirements:

If you did not receive any funding for this study, please simply state: “The authors received no specific funding for this work.

3. Please provide separate figure files in .tif or .eps format only and remove any figures embedded in your manuscript file. Please also ensure all files are under our size limit of 10MB.

4. In the online submission form, you indicated that "The datasets used in this study are available from the corresponding author on reasonable request". All PLOS journals now require all data underlying the findings described in their manuscript to be freely available to other researchers, either 1. In a public repository, 2. Within the manuscript itself, or 3. Uploaded as supplementary information.

Additional Editor Comments (if provided):

Reviewers' comments:

Reviewer's Responses to Questions

**Comments to the Author**

1. Does this manuscript meet PLOS Global Public Health’s publication criteria? Is the manuscript technically sound, and do the data support the conclusions? The manuscript must describe methodologically and ethically rigorous research with conclusions that are appropriately drawn based on the data presented.

Reviewer #1: Yes

Reviewer #2: No

Reviewer #3: Partly

2. Has the statistical analysis been performed appropriately and rigorously?

Reviewer #1: Yes

Reviewer #2: No

Reviewer #3: No

3. Have the authors made all data underlying the findings in their manuscript fully available (please refer to the Data Availability Statement at the start of the manuscript PDF file)?

Reviewer #1: Yes

Reviewer #2: No

Reviewer #3: Yes

4. Is the manuscript presented in an intelligible fashion and written in standard English?

Reviewer #1: Yes

Reviewer #2: No

Reviewer #3: No

5. Review Comments to the Author

Reviewer #1: Overall evaluation:

This is an interesting pilot study investigating the experiences and challenges faced by patients with noncommunicable diseases (NCDs) in Nepal during the COVID-19 pandemic. The study uses a mixed-methods approach combining both quantitative survey data and qualitative interviews to explore healthcare service availability and barriers to care for NCD patients.

The importance of the research question is clear - NCD patients are at higher risk for poor COVID-19 outcomes, yet little is known about how the pandemic impacted their healthcare access and disease management, especially in low-resource settings like Nepal. The objectives are specific, and the mixed-methods approach is appropriate. The sample sizes seem reasonable for an initial pilot study. The analyses utilize appropriate statistical tests and qualitative methods. The discussion situates the findings well within the larger literature and considers implications and future directions.

Overall, this manuscript makes a meaningful contribution to understanding NCD care amidst the COVID pandemic. I recommend minor revisions to strengthen the presentation and highlight key results prior to acceptance for publication.

Major comments:

1. The abstract and introduction could more clearly distinguish the study's novel contributions from what is already known about COVID's impacts on global NCD care per WHO reports.

2. Consider restructuring parts of the results section for clarity and concision:

- Combine results for Objectives 1 and 2 which both rely on the qualitative data

- Trim the MCA, HCA, and cluster details that do not directly address the research questions (perhaps move some details to supplement)

- Prioritize highlighting key survey results on healthcare access challenges

3. The discussion could comment further on:

- why access barriers existed when services were reportedly still available

- the finding that only 1 participant used telehealth

- comparisons to NCD impacts observed in other low- and middle-income countries

4. Address generalizability limitations: convenience sampling from only 2 districts

Minor comments:

Abstract

1. Outline the specific methods in brief (quant survey and qualitative interviews)

2. The conclusion could more directly connect back to the study objectives and purpose

Introduction

1. References 6 and 14 seem potentially irrelevant for context specific to Nepal - consider replacing with local prevalence/burden data if appropriate

Methods

1. Were interviews conducted with any of the same patients included in the quantitative sample? Clarify whether qualitative and quant participants overlapped.

2. Verify whether verbal or written consent was obtained (text seems contradictory)

Results

1. Consider organizing the qualitative results by the key themes in barriers/facilitators

2. For Fig 1: define "variables” analyzed in MCA plot (questions from survey?)

Discussion

1. In limitations: Note social desirability bias for phone interviews specifically

Tables/Figures - Well designed to support key results

References - Comprehensive and adequate

In summary, I recommend minor revision to better highlight the key findings before final acceptance. This pilot study makes a valuable contribution towards understanding NCD care access issues amidst COVID-19 in Nepal's resource-limited setting. Addressing the clarity/concision of results and highlighting novel contributions will further strengthen the manuscript for publication. Please feel free to contact me if you have any other questions!

Reviewer #2: The study involved collecting questionnaire responses from NCD patients pertaining to their situation and concerns during COVID-19. The authors devised an approach using MCA and HCA as well as qualitative analysis. The study identified many important situations that demands attention for remedies in the event of next pandemic. The results from the questionnaire is found to be insightful and the paper presents useful measures for pandemic preparedness. However, in its present form, it cannot be accepted due to the weakness found in using MCA. Furthermore, it requires language editing and clarifications as follows.

1) Line 86, did reference 15 report on the challenges and behavior of the people? Include relevant literature that also performed similar studies. Can references of similar studies of conditions faced by people with or without NCD during COVID-19 be provided? The paper is lacking of literature review of studies of understanding the situation of patience with illnessess or NCD during COVID-19.

2) Line 110, objective 1, why is there a special mention of diabetes? Is there a particular focus on diabetes and why? Please clarify.

3) Line 121, Clarify objective 2, "to the use of diabetes", what do you mean by that?

4) Line 139, remove the word purposively.

5) Line 167 - 199, it was not mentioned why these methods of analysis were chosen and in what way will these methods satisfy the aim of the study. There are many other methods which could also do the same analysis.

6) What were similar studies that used the approaches used in this paper? Why are these approaches considered suitable? Please explain.

7) Table 1, Provide number references to the questions in Table 1 for easy reference including the relevant questions reference numbers in the qualitative part of the study.

8) What does the * mean in Table 1 in MCA column?

9) What does it signify when the no. of patients in MCA is different from actual? Is there a bias in MCA towards categories with more samples e.g in Religion?

10) Line 252, The cummulative variance that indicates how much of the data is explained in the plot is less than 30%. Please show the variance and cummulative variance of each dimension. How can you justify or verify that what is reported regarding the contributing features to the clusters is true? This indicate that MCA biplot is not reliable and is not suitable analysis for to study the data.

11) Please provide meaningful subheadings instead of Objective 1, 2, etc.

12) Line 253, Please show the dendogram for the HCA so that we can see the hierarchy of the clustering and the cut-off where the 4 clusters are found. It is unclear why 4 groups are most suitable.

13) Please align the respective objectives with the questionnaire questions listed in Table 1 so that we can see the prevalence of that response.

E.g People's fear of getting infected by health workers was drawn from the question avoided in-person contact with high risk people or its question reference number.

Or if such objectives can be included in Table 1, for us to easily understand the prevalence of such behaviors

14) Can the study of objectives sections be more structured and objective? How many patients reported fear of getting infected, as well as the other potential barriers and what are the demographics of such patients? How many are from Kavrepalanchok and Nuwakot? How many facing which barriers are having which NCD?

It is unclear the benefits of showing the transcripts of the recordings. Please justify the benefits in the introduction as well as elaborate on the reason for presenting only the recordings and not other statistical analysis of the survey with regards to this objective, apart from the clustering.

15) How can you verify that the MCA and HCA results are the correct or best ones to present?

16) Line 382, while the authors claim that data mining methods were used to obtain as much as possible, however the recordings presented in the objective sections can be mined for further analysis.

17) Line 471, the negative feelings evoked by social media were stronger in patients who were concerned about COVID-19 and ...... than who? Please specify, it is unclear what is being compared here.

18) Line 491, please rephr

---

## [Decision Letter · Decision Letter 1]

2 Jul 2024

Exploring ways to support patients with noncommunicable diseases: a pilot study in Nepal during the COVID-19 pandemic

PGPH-D-23-02128R1

Dear Dr. Iwashita,

We are pleased to inform you that your manuscript 'Exploring ways to support patients with noncommunicable diseases: a pilot study in Nepal during the COVID-19 pandemic' has been provisionally accepted for publication in PLOS Global Public Health.

Best regards,

Sok King Ong

Academic Editor

Reviewer Comments (if any, and for reference):

Reviewer's Responses to Questions

**Comments to the Author**

1. If the authors have adequately addressed your comments raised in a previous round of review and you feel that this manuscript is now acceptable for publication, you may indicate that here to bypass the “Comments to the Author” section, enter your conflict of interest statement in the “Confidential to Editor” section, and submit your "Accept" recommendation.

Reviewer #1: All comments have been addressed

Reviewer #2: All comments have been addressed

2. Does this manuscript meet PLOS Global Public Health’s publication criteria? Is the manuscript technically sound, and do the data support the conclusions? The manuscript must describe methodologically and ethically rigorous research with conclusions that are appropriately drawn based on the data presented.

Reviewer #1: Yes

Reviewer #2: Yes

3. Has the statistical analysis been performed appropriately and rigorously?

Reviewer #1: Yes

Reviewer #2: Yes

4. Have the authors made all data underlying the findings in their manuscript fully available (please refer to the Data Availability Statement at the start of the manuscript PDF file)?

Reviewer #1: Yes

Reviewer #2: Yes

5. Is the manuscript presented in an intelligible fashion and written in standard English?

Reviewer #1: Yes

Reviewer #2: Yes

6. Review Comments to the Author

Reviewer #1: Thanks for addressing all the comments.

Reviewer #2: The authors have made great effort to improve the manuscript, addressing all comments satisfactorily.

The manuscript is ready for publication.

7. PLOS authors have the option to publish the peer review history of their article (what does this mean?). If published, this will include your full peer review and any attached files.

**Do you want your identity to be public for this peer review?** For information about this choice, including consent withdrawal, please see our Privacy Policy.

Reviewer #1: **Yes: **Abdul Hanif Khan Yusof Khan

Reviewer #2: No
